# Confidence Intervals and Simultaneous Confidence Bands Based on Deep Learning

**Asaf Ben Arie**                                                    *asafbenarie@mail.tau.ac.il*
*Department of Statistics and Operations Research*
*Tel Aviv University, Israel*

**Malka Gorfine**                                                    *gorfinem@tauex.tau.ac.il*
*Department of Statistics and Operations Research*
*Tel Aviv University, Israel*

**Reviewed on OpenReview:** *https: // openreview. net/ forum? id=XXXX*

## Abstract

Deep learning models have significantly improved prediction accuracy in various fields, gaining recognition across numerous disciplines. Yet, an aspect of deep learning that remains insufficiently addressed is the assessment of prediction uncertainty. Producing reliable uncertainty estimators could be crucial in practical terms. For instance, predictions associated with a high degree of uncertainty could be sent for further evaluation. Recent works in uncertainty quantification of deep learning predictions, including Bayesian posterior credible intervals and a frequentist confidence-interval estimation, have proven to yield either invalid or overly conservative intervals. Furthermore, there is currently no method for quantifying uncertainty that can accommodate deep neural networks for survival (time-to-event) data that involves right-censored outcomes. In this work, we provide a non-parametric bootstrap method that disentangles data uncertainty from the noise inherent in the adopted optimization algorithm. The validity of the proposed approach is demonstrated through an extensive simulation study, which shows that the method is accurate (i.e., valid and not overly conservative) as long as the network is sufficiently deep to ensure that the estimators provided by the deep neural network exhibit minimal bias. Otherwise, undercoverage of up to 8% is observed. The proposed ad-hoc method can be easily integrated into any deep neural network without interfering with the training process. The utility of the proposed approach is demonstrated through two applications: constructing simultaneous confidence bands for survival curves generated by deep neural networks dealing with right-censored survival data, and constructing a confidence interval for classification probabilities in the context of binary classification regression. Code for the data analysis and reported simulation is available at Githubsite: `https://github.com/Asafba123/Survival_bootstrap`.

## 1 Introduction

Deep neural networks (DNNs) have gained popularity due to several key factors: (i) High performance - DNNs have demonstrated superior performance in various tasks, such as image recognition, speech processing, and natural language understanding, often surpassing traditional statistical or machine learning methods. (ii) Ability to learn complex patterns - with multiple layers of neurons, DNNs can learn intricate and hierarchical patterns in data, making them effective for tasks that involve complex data structures. (iii) Scalability - DNNs can handle large-scale data and benefit from the availability of big data and powerful computational resources. (iv) Flexibility and adaptability - DNNs can be adapted to a wide range of applications and domains. Ongoing research and development in neural network architectures, optimization techniques, and hardware accelerators, such as GPUs and TPUs, continually enhance the capabilities and efficiency of DNNs.

These factors, among others, have led to the widespread use and popularity of deep neural networks in both academia and industry (LeCun et al., 2015).

However, a recent survey on uncertainty in DNN (Gawlikowski et al., 2023) suggests that the real-world applications of DNNs are still restricted, primarily because of their failure to offer reliable estimates of uncertainty. Accurate uncertainty estimates are of high practical importance. For instance, predictions made with a high degree of uncertainty can be either disregarded or referred to human specialists for further evaluation (Gal & Ghahramani, 2016; Gawlikowski et al., 2023, and references therein). In this work, we present a resampling-based estimator of DNN prediction uncertainty that can be integrated with any DNN algorithm without interfering with its operation or compromising its accuracy.

## 1.1 Quantifying Uncertainty in Deep Learning - Related Works

Recently, there has been increasing attention to assessing uncertainty within DNNs. Gawlikowski et al. (2023) provided a comprehensive overview of uncertainty estimation in neural networks, reviewing recent advances in the field and categorizing existing methods into four groups based on the number (single or multiple) and the nature (deterministic or stochastic) of the neural networks used. Alaa & Van Der Schaar (2020) noted that existing methods for uncertainty quantification predominantly rely on Bayesian neural networks (BNNs), while Bayesian credible intervals do not guarantee frequentist coverage, and approximate posterior inference undermines discriminative accuracy. They also mentioned that BNNs often require major modifications to the training procedure, and exact Bayesian inference is computationally prohibitive in practice. Moreover, Kuleshov et al. (2018) and Alaa & Van Der Schaar (2020) demonstrated that Bayesian uncertainty estimates often fail to capture the true data distribution. For instance, a 95% posterior credible interval generally contains the true value much less than 95% of the time. Kuleshov et al. (2018) suggested an additional calibration step, showing that it can guarantee the desired coverage rate for a variety of BNN regression algorithms.

Alaa & Van Der Schaar (2020) introduced a discriminative jackknife (DJ) procedure for point-wise predictive confidence intervals, a frequentist method based on Barber et al. (2021). Their approach is applicable to various DNN models, easy to implement, and can be applied in an ad-hoc fashion without interfering with model training or compromising its accuracy. The experiments in their study show that DJ is often overly conservative, with coverage rates exceeding the desired nominal level. For example, achieving a 100% coverage rate instead of the intended 90%. This conservatism, while valid, suggests that narrower intervals could be used while still maintaining the desired nominal coverage level. Obviously, using a narrower confidence interval at the same confidence level is more informative and thus advantageous.

The resampling method presented in the current work will be demonstrated to provide an accurate coverage rate, as long as the estimators provided by the DNN have only a small bias. In such cases, our proposed approach effectively achieves the desired target coverage rate without being overly conservative. While the point-wise predictive confidence interval by Alaa & Van Der Schaar (2020) is suitable for various deep learning models, it is not applicable to deep learning for survival analysis for two reasons. First, their approach relies on residuals—-the difference between the actual and predicted outcomes—-but in censored survival data, the actual outcome is often unobservable. Second, they focus on confidence intervals for the outcome of a new observation, whereas in some deep learning applications, it is more relevant to estimate a function of a new observation, such as the probability of a new patient surviving the next ten years. This limitation will be elaborated further in the following sections. Our proposed approach is suitable for various deep learning methods, including those used in survival analysis.

## 1.2 Survival Prediction with Deep Learning

Survival analysis involves modeling the time until a predefined event occurs. This type of analysis is common in various fields such as medicine, public health, epidemiology, engineering, and finance. Survival prediction methods are often used at some "baseline" time, such as the time of diagnosis or treatment initiation, to address questions such as, "What is the probability that this individual will be alive in ten years, given their baseline characteristics?" Clinicians often use these predicted probabilities to make critical decisions about patient care and the implementation of specific therapies. In the context of credit risk, a credit scoring model

involves predicting the probability that an account will default over a future time period based on a number of observed features that characterize account holders or applicants.

Training datsets of survival data typically include censored or truncated observations. Censored data arise when the exact time of the event is unknown but falls within a certain period. There are several types of censoring. *Right censoring:* The individual is known to be free of the event at a given time. *Left censoring:* The individual experienced the event before the study began. *Interval censoring:* The event occurs within a specified interval. Truncation involves different schemes. *Left truncation:* Only individuals with event time beyond a certain time are included in the sample. *Right truncation:* Only individuals who have experienced the event by a specific time are included. Each type of censoring and truncation requires a specialized estimation procedure to obtain reliable risk predictions (Klein & Moeschberger, 2006; Kalbfleisch & Prentice, 2011). In this work, we mainly focus on right-censored data since this is the most common setting. However, our approach is general and can be adopted to any type of censoring and truncation.

The popularity of survival DNNs has been on the rise in recent years. Nearly every new development in DNN is swiftly accompanied by an adaptation for survival analysis. Recent reviews on deep learning for survival analysis can be found in Hao et al. (2021); Zhong et al. (2022); Deepa & Gunavathi (2022); Wiegrebe et al. (2024). To summarize, Faraggi & Simon (1995) replaced the linear component of the well-known Cox proportional hazards model Cox (1972) with a one-hidden-layer neural network. However, their approach did not produce significant improvements over the traditional Cox model in terms of the concordance index Harrell et al. (1982). Later, Katzman et al. (2016) expanded on Faraggi & Simon (1995)'s approach by using a multilayer neural network, named DeepSurv, which achieved remarkable results in a study on breast cancer. Various adaptations of the Cox model using more complex architectures have been developed for specific applications, such as genomics data, clinical research, and medical imaging (Yousefi et al., 2017; Ching et al., 2018; Matsuo et al., 2019; Haarburger et al., 2019; Li et al., 2019). Additional DNNs methods for survival data include Ranganath et al. (2016); Chapfuwa et al. (2018); Giunchiglia et al. (2018); Lee et al. (2018); Ren et al. (2019), among others. In the comprehensive systematic review by Wiegrebe et al. (2024), it is observed that of the 61 reviewed methods, 26 are based on extended versions of the Cox model. This includes DeepSurv by Katzman et al. (2016) and CoxTime by Kvamme et al. (2019), an efficient and flexible extension of DeepSurv that incorporates time-dependent effects. Another popular approach is the DeepHit of Lee et al. (2018), a discrete-time deep learning survival method that avoids assumptions about the underlying stochastic process. However, all current DNN approaches for survival analysis overlook estimating the uncertainty of predictions.

## 1.3 Contributions

In this work, we introduce a novel resampling approach for estimating prediction uncertainty, applicable to a wide range of deep neural networks and machine learning methods. The approach is presented in a general context, with its utility demonstrated through two applications: survival analysis and binary classification regression.

Point-wise confidence intervals and simultaneous confidence bands are used to express the uncertainty around estimated parameters or functions, but they differ in their scope and interpretation. In survival analysis, for example, a point-wise confidence interval for the survival probability at a certain time point is only valid for that single time point. A simultaneous confidence band provides an estimate for the value of a function over a range of points. Confidence bands are generally more complex to construct due to the need to account for the correlation between points over the range. A common incorrect practice is to plot point-wise confidence intervals for multiple time points and to interpret the resulting curves as a confidence band. This misinterpretation suggests, for example, 95% confidence that the area between the curves covers the true survival curve. However, it is well known that the bands obtained this way are too narrow for such an inference to be correct. To the best of our knowledge, no published works exist on confidence bands based on neural network or DNN analysis. Moreover, the currently existing methods for prediction uncertainty based on DNNs are not applicable for survival data with censoring.

In this work, we present a bootstrap approach that can be applied ad hoc to any deep neural network (DNN) for estimating uncertainty. Unlike existing methods, our approach provides: (1) Valid point-wise confidence

intervals that are not overly conservative, provided the estimators generated by the DNN exhibit minimal bias. Otherwise, undercoverage may occur. (2) The first simultaneous confidence bands specifically for DNNs. (3) The first point-wise confidence intervals and confidence bands tailored for survival DNNs.

## 2 Bootstrap with DNN and the Proposed Bootstrap Approach

### 2.1 Setup

Optimization of any DNN usually involves multiple random steps, such as initial values, batch partitions, and training-validation data splitting (the training set is used for computing the loss function, and the validation set is used for stopping criteria in the optimization procedure). Running the same DNN on the same dataset multiple times provides highly similar results but not necessarily identical ones. Some of the random steps, such as the step of training-validation data splitting, can be easily controlled by the users without compromising the optimization quality, but others, such as batch partitions, cannot.

A naive application of a non-parametric bootstrap for prediction uncertainty estimation involves training multiple DNNs on different bootstrap training samples, which are random samples with replacement from the training dataset. One of the fundamental requirements for successfully applying the bootstrap procedure is to apply the exact same procedure on each bootstrap sample as done on the original training data. However, in DNN procedures, bootstrap samples will often be based on nonidentical random steps that are intrinsic parts of the DNN and cannot easily be controlled by the user or are not recommended to be fixed. Hence, the results of the bootstrap samples include not only variability due to the data but also arbitrary variability due to the use of different randomness in various steps within the optimization procedure. Therefore, such a naive approach is expected to overestimate uncertainty, as will be shown below.

Let $\widehat{\theta}_n$ be the DNN estimator based on $n$ observations (training and validation), to be used for prediction or estimation at a new test point. It is desirable for a prediction or estimation to be accompanied by a measure of error or uncertainty. In particular, we consider an interval or band around the prediction or estimation, as defined in the following examples.

**Example 1:** Consider a standard supervised learning setting where $\mathbf{x}$ denotes the features, $\mathbf{x} \in \mathcal{X} \subseteq \mathbb{R}^d$, and $y \in \mathcal{Y}$ denotes the outcome. A prediction model $f(\mathbf{x}; \theta)$ is trained to predict $y$ using the data $\mathcal{D}_n = \{(\mathbf{x}_i, y_i), i = 1, \ldots, n\}$ and we get $f(\mathbf{x}; \widehat{\theta}_n)$. For a new test point with features $\mathbf{x} \in \mathcal{X}$ the outcome $y$ is predicted by $f(\mathbf{x}; \widehat{\theta}_n)$. A point-wise confidence interval (also known as a prediction interval) $[L(\mathbf{x}), U(\mathbf{x})]$ of level $(1 - \alpha)100\%$, $\alpha \in (0, 1)$, is defined by

$$\Pr\{y \in [L(\mathbf{x}), U(\mathbf{x})]\} = 1 - \alpha \tag{1}$$

where $L(\mathbf{x})$ and $U(\mathbf{x})$ are functions of $f(\mathbf{x}; \widehat{\theta}_n)$ and the probability is taken with respect to a new test point $(\mathbf{x}, y)$ and $\mathcal{D}_n$. An estimator of uncertainty in the prediction of $y$ is expressed through the pointwise confidence interval $[L(\mathbf{x}), U(\mathbf{x})]$. For any given $\alpha$, it is desired to find the pair $L(\mathbf{x})$ and $U(\mathbf{x})$ that will be close together in some sense. For example, if $U(\mathbf{x}) - L(\mathbf{x})$, which represents the length of the confidence interval, is random, the goal is to select the pair $L(\mathbf{x})$ and $U(\mathbf{x})$ that makes the average length of the interval smallest.

Alaa & Van Der Schaar (2020) considered this example but with a conservative definition of the point-wise confidence interval, such that the equality sign in Eq. (1) is replaced with $\geq$. Therefore, their proposed approach is often provides a confidence level much higher than $1 - \alpha$ at the cost of a wider interval. Clearly, wider intervals imply less confidence, and vice versa.

**Example 2:** In time-to-event data with right censoring, the true event times are typically unknown for all individuals. The training and validation data consists of $y_i = (t_i, d_i)$ and $\mathbf{x}_i$ for each observation $i$, where $t_i = \min(t_i^*, c_i)$, $t_i^*$ and $c_i$ are the event and right-censoring times, respectively, and $d_i = I(t_i = t_i^*)$ is the event indicator. Hence, the dataset (training and validation) consists of $\mathcal{D}_n = \{(\mathbf{x}_i, t_i, d_i), i = 1, \ldots, n\}$. For a new test point $\mathbf{x}$, the goal in survival analysis is usually not to predict the event time $t^*$ based on $\mathbf{x}$, but rather to provide an estimator for the conditional survival function, given $\mathbf{x}$, within a pre-specified interval,

e.g. $[0, \tau]$, $\tau > 0$. Specifically, the conditional survival function is defined by

$$S(s|\mathbf{x}) = \Pr(t^* > s|\mathbf{x}; \theta) \ , \mathbf{x} \in \mathcal{X} \ , s \in [0, \infty) \, .$$

Hence, in this setting, $\widehat{S}_n(s|\mathbf{x}, \widehat{\theta}_n)$ is an estimated survival *curve* produced by the DNN (e.g., DeepHit and CoxTime) for a vector of features $\mathbf{x}$ and $s \in [0, \tau]$. Usually, $\tau$ is defined not only by the desired upper value but also by the data. For example, the survival function cannot be reliably estimated beyond the last observed failure time (i.e., the maximum value of $t_i$ such that $d_i = 1$).

A point-wise confidence interval for a new test point $\mathbf{x}$ provides a confidence interval for the survival function at a fixed time $s_0 \in [0, \tau]$ and is given by

$$\Pr\{S(s_0|\mathbf{x}) \in [L(s_0|\mathbf{x}), U(s_0|\mathbf{x})]\} = 1 - \alpha \, , \tag{2}$$

where $L(s_0|\mathbf{x})$ and $U(s_0|\mathbf{x})$ are functions of $\widehat{S}_n(s_0|\mathbf{x}, \widehat{\theta}_n)$ and the probability is taken with respect to a new test point $\mathbf{x}$ and $\mathcal{D}_n$. In many applications (Sachs et al., 2022, and references therein), it is of interest to find upper and lower *curves* which guarantee, with a given confidence level, that the conditional survival function of a new test $\mathbf{x}$ is covered within the curves for all $s$ in a desired interval $\mathcal{B}$, $\mathcal{B} \subseteq [0, \tau]$. Namely, we wish to find two random functions $L(s|\mathbf{x})$ and $U(s|\mathbf{x})$, which are functions of $\widehat{S}_n(s|\mathbf{x}, \widehat{\theta}_n)$, such that

$$\Pr\{S(s|\mathbf{x}) \in [L(s|\mathbf{x}), U(s|\mathbf{x})], \ \text{for all} \ s \in \mathcal{B}\} = 1 - \alpha \, , \tag{3}$$

and the probability is taken with respect to a new test point $\mathbf{x}$ and $\mathcal{D}_n$. We call such a pair of curves $L(s|\mathbf{x})$ and $U(s|\mathbf{x})$, $s \in \mathcal{B}$, a $(1 - \alpha)100\%$ simultaneous confidence band for $S(s|\mathbf{x})$, $s \in \mathcal{B}$.

In the next section, we present a very general resampling-based approach for constructing confidence intervals and simultaneous confidence bands.

## 2.2 Main Idea - Ensemble-Based Bootstrap Procedure

In the following, we develop a bootstrap approach as a method of estimating the distribution of a function of a DNN statistic and a new data point, denoted by $\widehat{\mu}_n(\mathbf{x}, \widehat{\theta}_n)$. In examples 1 and 2 above, $\widehat{\mu}_n(\mathbf{x}, \widehat{\theta}_n) = f(\mathbf{x}; \widehat{\theta}_n)$ and $\widehat{\mu}_n(\mathbf{x}, \widehat{\theta}_n) = \widehat{S}_n(s_0|\mathbf{x}, \widehat{\theta}_n)$, respectively. To generate valid confidence intervals or simultaneous confidence bands based on DNNs, it is required to assume that the DNNs under consideration provide consistent estimators (Carney et al., 1999; Alaa & Van Der Schaar, 2020). Although these assumptions do not hold in general, it is often accepted that the bias of various DNNs estimators is small and much smaller than the variance (Geman et al., 1992; Carney et al., 1999). The simulation study in Sections 3 and 4 supports the assumptions regarding CoxTime (Kvamme et al., 2019) and classification regression with deep networks, under certain smoothness constraints, given a sufficiently large training dataset and a sufficiently deep network.

Let $\mu^o$ denote the true value of the desired quantity and assume that the distribution of the difference, $\widehat{\mu}_n - \mu^o$, contains all the information needed for assessing the precision of $\widehat{\mu}_n$. We start by decomposing $\widehat{\mu}_n$ into a sum of two random variables $Z_1$, $Z_2$, such that

$$Z_1 - \mu^o \sim F_1(0, v_1) \ \text{and} \ Z_2 \sim F_2(0, v_2) \tag{4}$$

where $F_1$ and $F_2$ are unknown distributions with means 0 and variances $v_1$ and $v_2$, respectively, so the desired variance is

$$var(\widehat{\mu}_n) = v_1 + v_2 + 2v_{12}$$

where $v_{12} = cov(Z_1, Z_2)$. $Z_1$ is constant given the dataset $\mathcal{D}_n$, while $Z_2$ captures the added variability due to the inherited randomness of the DNN optimization procedure. Thus, $Z_2$ is not constant given $\mathcal{D}_n$, and as long as $v_2 > 0$, running the same DNN on the same dataset multiple times provides highly similar results but not necessarily identical ones. A naive bootstrap procedure based on $B$ bootstrap samples of the training dataset provides $\widehat{\theta}_n^{(b)}$ and then $\widehat{\mu}_n^{(b)} = \widehat{\mu}_n^{(b)}(\mathbf{x}, \widehat{\theta}_n^{(b)})$, $b = 1, \ldots, B$. Similarly, each $\widehat{\mu}_n^{(b)}$ is decomposed into $Z_1^{(b)} + Z_2^{(b)}$ where the conditional distributions, given $\mathcal{D}_n$, are

$$Z_1^{(b)} - Z_1|\mathcal{D}_n \sim F_1(0, v_1) \ \text{and} \ Z_2^{(b)}|\mathcal{D}_n \sim F_2(0, v_2) \, . \tag{5}$$

Therefore,

$$var\left(Z_1^{(b)} + Z_2^{(b)} - Z_1 - Z_2|\mathcal{D}_n\right) = v_1 + 2v_2 + 2v_{12} \tag{6}$$

since $cov(Z_j^{(b)}, Z_2|\mathcal{D}_n) = 0$, $j = 1, 2$, $b = 1, \ldots, B$. The extra variance, $v_2$, in Eq.(6), is the reason that a naive bootstrap variance estimator, overestimates the variance. However, even if the naive bootstrap procedure provides conservative confidence intervals or simultaneous confidence bands, it should not generally be considered a valid approach, as it includes arbitrary noise that is not part of the true uncertainty we wish to estimate. This will be demonstrated in the simulation study in Sections 3 and 4.

The above decomposition motivates the following ensemble-based bootstrap procedure for eliminating the extra variance $v_2$:

1. Generate an ensemble estimator $\widehat{\mu}_n^M = M^{-1}\sum_{m=1}^M \widehat{\mu}_{n,m}$, where $\widehat{\mu}_{n,m} = \widehat{\mu}_{n,m}(\mathbf{x}, \widehat{\theta}_{n,m})$ and $\widehat{\theta}_{n,1}, \ldots, \widehat{\theta}_{n,M}$ are $M$ repetitions of the DNN applied on the original dataset without interfering with the model training. $M$ should be large enough such that $v_2/M \approx 0$. In our simulation and data analysis we found that $M = 100$ is sufficient.

2. Evaluate the bootstrap distribution based on $\widehat{\mu}_n^{(1)}, \ldots, \widehat{\mu}_n^{(B)}$ around $\widehat{\mu}_n^M$ (instead of $\widehat{\mu}_n$).

This approach requires $B + M$ runs of the DNN.

A DNN training process is based on an iterative process with a convergence rule. To avoid over fitting, during training, the model is evaluated on a holdout validation dataset after each epoch, while the gradient descent procedure is applied only on the training dataset. If the performance of the model, evaluated on the validation dataset, is not improving for a pre-specified number of consecutive runs (known as patience), that is, the validation loss function increases, the training process is stopped. The final model is the best performing model in terms of validation loss. The patience is often set somewhere between 10 and 100 (10 or 20 is more common, we used 15), but it depends on the dataset and network. To avoid overlap between the training and holdout validation dataset within each bootstrap sample, we first do training-validation splitting and then bootstrap the training dataset.

Consider, for example, a binary classification regression where $y \in \{0, 1\}$. The objective is to estimate the probability $p(\mathbf{x}) = \Pr(y = 1|\mathbf{x})$ using a DNN estimator, $\widehat{\mu}_n = \widehat{p}_n(\mathbf{x})$, along with the corresponding confidence interval. Specifically, we aim to satisfy

$$\Pr\left(|\widehat{p}_n(\mathbf{x}) - p(\mathbf{x})| \leq d(\mathbf{x})\right) = 1 - \alpha. \tag{7}$$

The proposed ensemble-based bootstrap variance estimator of $\widehat{\mu}_n$ is given by $B^{-1}\sum_{b=1}^B (\widehat{\mu}_n^{(b)} - \widehat{\mu}_n^M)^2$, where $\widehat{\mu}_n^M = \widehat{p}_n^M(\mathbf{x}) = M^{-1}\sum_{m=1}^M \widehat{p}_{n,m}(\mathbf{x})$. The corresponding ensemble-based confidence interval $[L(\mathbf{x}), U(\mathbf{x})]$ is constructed through the following steps:

1. For each bootstrap sample $b$, $b = 1, \ldots, B$, get $d^{(b)}(\mathbf{x}) = |\widehat{p}_n^{(b)}(\mathbf{x}) - \widehat{p}_n^M(\mathbf{x})|$.

2. Get the $1 - \alpha$ percentile of $d^{(b)}(\mathbf{x})$, $b = 1, \ldots, B$, denoted as $d_{1-\alpha}^{boots}(\mathbf{x})$.

3. Define $L(\mathbf{x}) = \max\{\widehat{p}_n(\mathbf{x}) - d_{1-\alpha}^{boots}(\mathbf{x}), 0\}$ and $U(\mathbf{x}) = \min\{\widehat{p}_n(\mathbf{x}) + d_{1-\alpha}^{boots}(\mathbf{x}), 1\}$.

In the next subsection we demonstrate the proposed ensemble-bootstrap for constructing simultaneous confidence band for $S(s|\mathbf{x})$ of Example 2.

## 2.3 Simultaneous Confidence Bands

We start with a simultaneous confidence bands for the survival function based on the well-known Kolmogorov-Smirnov test statistics (Mood et al., 1963). For a new $\mathbf{x}$, we look for a constant $d(\mathbf{x})$ (constant with respect to $s$) such that

$$\Pr\left(\sup_{s \in [0, \tau]} \left|\widehat{S}_n(s|\mathbf{x}; \widehat{\theta}_n) - S(s|\mathbf{x})\right| \leq d(\mathbf{x})\right) = 1 - \alpha,$$

and $d(\mathbf{x})$ is computed based on the ensemble-bootstrap procedure. The following is the complete algorithm for generating a simultaneous confidence band of $S(\cdot|\mathbf{x})$ at level of $(1-\alpha)100\%$:

1. Generate an ensemble estimator, for example, $\widehat{S}_n^M(s|\mathbf{x}) = M^{-1}\sum_{m=1}^M \widehat{S}_{n,m}(s|\mathbf{x},\widehat{\theta}_{n,m})$, $s \in [0,\tau]$.

2. For any bootstrap sample $b$, $b = 1,\ldots,B$, get $\widehat{S}_n^{(b)}(s|\mathbf{x};\widehat{\theta}_n^{(b)}) = \widehat{\mu}_n^{(b)}(\mathbf{x},\widehat{\theta}_n^{(b)})$, $s \in [0,\tau]$, and

$$d^{(b)}(\mathbf{x}) = \max_{s\in[0,\tau]} \left| \widehat{S}_n^{(b)}(s|\mathbf{x};\widehat{\theta}_n^{(b)}) - \widehat{S}_n^M(s|\mathbf{x}) \right|.$$

3. Get the $1-\alpha$ percentile of $d^{(1)}(\mathbf{x}),\ldots,d^{(B)}(\mathbf{x})$, denoted by $d_{1-\alpha}^{boots}(\mathbf{x})$.

4. For any $s \in [0,\tau]$, we define

$$L^{KS}(s|\mathbf{x}) = \max\left\{ \widehat{S}_n(s|\mathbf{x};\widehat{\theta}_n) - d_{1-\alpha}^{boots}(\mathbf{x}),\, 0 \right\}$$

and

$$U^{KS}(s|\mathbf{x}) = \min\left\{ \widehat{S}_n(s|\mathbf{x};\widehat{\theta}_n) + d_{1-\alpha}^{boots}(\mathbf{x}),\, 1 \right\}.$$

The above $L^{KS}$ and $U^{KS}$ provide a fixed-width band. The following weight-based bands have varied widths as a function of $s$, $s \in \mathcal{B}$, and are expected to provide narrower bands. Specifically, for a new $\mathbf{x}$, we now look for $d^p(\mathbf{x})$, such that

$$\Pr\left( \sup_{s\in[0,\tau]} \frac{\left| \widehat{S}_n(s|\mathbf{x};\widehat{\theta}_n) - S(s|\mathbf{x}) \right|}{\widehat{W}_n(s|\mathbf{x};\widehat{\theta}_n)} \le d^p(\mathbf{x}) \right) = 1-\alpha,$$

where

$$\widehat{W}_n(s|\mathbf{x};\widehat{\theta}_n) = [\widehat{S}_n^*(s|\mathbf{x};\widehat{\theta}_n)\{1 - \widehat{S}_n^*(s|\mathbf{x};\widehat{\theta}_n)\}]^{\frac{1}{2}}$$

and $\widehat{S}_n^*(s|\mathbf{x};\widehat{\theta}_n)$ is defined similarly to $\widehat{S}_n(s|\mathbf{x};\widehat{\theta}_n)$ but with values truncated to the range of 0.01 to 0.99. Hence, the width of the confidence band of $S(s|\mathbf{x})$ based on the above, is proportional to the "approximated" standard error of its estimator, $\{\widehat{S}_n(s|\mathbf{x};\widehat{\theta}_n)(1 - \widehat{S}_n(s|\mathbf{x};\widehat{\theta}_n)\}^{\frac{1}{2}}$. The use of truncated survival estimator $\widehat{S}_n^*(s|\mathbf{x};\widehat{\theta}_n)$ in the denominator is in order to avoid instabilities in cases where $\widehat{S}_n(s|\mathbf{x};\widehat{\theta}_n)$ is too close to either 1 or 0. The following is the modified algorithm:

1. Generate an ensemble estimator, for example, $\widehat{S}_n^M(s|\mathbf{x}) = M^{-1}\sum_{m=1}^M \widehat{S}_{n,m}(s|\mathbf{x};\widehat{\theta}_{n,m})$, $s \in [0,\tau]$.

2. For any bootstrap sample $b$, $b = 1,\ldots,B$, get $\widehat{S}_n^{(b)}(s|\mathbf{x};\widehat{\theta}_n^{(b)})$, $s \in [0,\tau]$, and

$$d^{p,(b)}(\mathbf{x}) = \max_{s\in[0,\tau]} \frac{\left| \widehat{S}_n^{(b)}(s|\mathbf{x};\widehat{\theta}_n^{(b)}) - \widehat{S}_n^M(s|\mathbf{x}) \right|}{\widehat{W}_n^{(b)}(s|\mathbf{x};\widehat{\theta}_n^{(b)})}.$$

3. Get the $1-\alpha$ percentile of $d^{p,(1)}(\mathbf{x}),\ldots,d^{p,(B)}(\mathbf{x})$, denoted by $d_{1-\alpha}^{p,boots}(\mathbf{x})$.

4. For any $s \in [0,\tau]$, we define

$$L^{prop}(s|\mathbf{x}) = \max\left\{ \widehat{S}_n(s|\mathbf{x};\widehat{\theta}_n) - \widehat{W}_n(s|\mathbf{x};\widehat{\theta}_n)d_{p,1-\alpha}^{boots}(\mathbf{x}),\, 0 \right\}$$

and

$$U^{prop}(s|\mathbf{x}) = \min\left\{ \widehat{S}_n(s|\mathbf{x};\widehat{\theta}_n) + \widehat{W}_n(s|\mathbf{x};\widehat{\theta}_n)d_{p,1-\alpha}^{boots}(\mathbf{x}),\, 1 \right\}.$$

Through a comprehensive simulation study and benchmark datasets, we show in the next sections that confidence bands based on the naive bootstrap approach are overly conservative, resulting in wide confidence bands. In contrast, with deep networks and a sufficiently large training datasets, the proposed approach provides the desired confidence level with narrower confidence widths compared to the naive bootstrap approach. Moreover, the weight-based bands, $L^{prop}$ and $U^{prop}$, are narrower than those of $L^{KS}$ and $U^{KS}$.

# 3 Simulation Study - Survival Analysis

We illustrate the performance of the proposed ensemble-based approach by constructing confidence bands for survival curves, specifically focusing on the CoxTime DNN (Kvamme et al., 2019). When comparing CoxTime with DeepSurv (Katzman et al., 2016), the two leading DNNs methods for survival analysis, we observe that DeepSurv often shows substantial bias in estimating $S(\cdot|\mathbf{x})$ for new observations, whereas CoxTime shows substantially less bias.

CoxTime is highly efficient in terms of runtime. One of its effective shortcuts cleverly adopts concepts from nested case-control designs, utilizing a small random subsample of the risk set, referred to as controls, for each loss function evaluation at each observed failure time, instead of the entire risk set. Moreover, the controls are resampled at each epoch. Users can specify the number of controls. While the authors suggested that even a very small number of controls, as few as one, is sufficient, it will be demonstrated here that the number of controls can substantially influence the performance of confidence bands.

## 3.1 Data Generation and Measures of Performances

We evaluated five different settings, each with multiple repetitions. The extensive number of repetitions for each setting generated a significant amount of data. To save space, the results were stored and summarized using a grid of points $\mathcal{T}$ defined for each setting. The hazard functions considered in the simulation study are of a general form

$$h(t|\mathbf{x}) = h_0(t)\exp\{g(t,\mathbf{x})\},$$

with the functions $h_0$ and $g$ based on Kvamme et al. (2019) (Settings 1–3) and Zhong et al. (2022) (Settings 4–5):

**Setting 1** Linear proportional hazards (PH) model: $h_0(t) = 0.1$, $g(t,\mathbf{x}) = g(\mathbf{x}) = \beta^T\mathbf{x}$ with $\beta = (0.44, 0.66, 0.88)$. Each covariate $x_j$, $j = 1, 2, 3$, was uniformly sampled from $U[-1, 1]$. Censoring times were generated from an exponential distribution with parameter $1/30$, and individuals still at risk at time $t = 30$ were censored at that time, resulting in approximately a 30% censoring rate. $\mathcal{T} = [0.1, 27]$ with jumps of 0.1.

**Setting 2** Non linear PH model: here

$$g(\mathbf{x}) = \beta^T\mathbf{x} + 2/3(x_1^2 + x_3^2 + x_1x_2 + x_1x_3 + x_2x_3)$$

and the rest follows Setting 1. This resulted in approximately 20% censoring rate.

**Setting 3** Non linear and non PH model: $h_0(t) = 0.02$, $g(t,\mathbf{x}) = a(\mathbf{x}) + b(\mathbf{x})t$, $a(\mathbf{x}) = \beta^T\mathbf{x} + 2/3(x_1^2 + x_3^2 + x_1x_2 + x_1x_3 + x_2x_3) + x_3$, $b(\mathbf{x}) = \{0.2(x_0 + x_1) + 0.5x_0x_1\}^2$ and the rest follows Setting 1. This resulted in approximately 30% censoring rate.

**Setting 4** Non linear PH model: $h_0(t) = 0.1t$ and

$$g(\mathbf{x}) = x_1^2x_2^3 + \log(x^3 + 1) + \sqrt{x_4x_5 + 1} + \exp(x_5/2) - 8.2$$

were the covariates generated from a Gaussian copula on $[0, 2]$ and correlation parameter 0.5. Right-censoring times were generated from an exponential distribution with parameter $1/28$, and individuals still at risk at time $t = 34$ were censored at that time. This resulted in approximately a 50% censoring rate. $\mathcal{T} = [2, 34]$ with jumps of 0.1.

**Setting 5** Non linear PH model: here $g(\mathbf{x})$ is replaced by

$$g(\mathbf{x}) = \{x_1^2x_2^3 + \log(x_3 + 1) + \sqrt{x_4x_5 + 1} + \exp(x_5/2)\}^2/20 - 6.0,$$

and the censoring parameter equals $1/45$. The rest follows Setting 4. This resulted in approximately 60% censoring rate.

Some of the above settings differ from the source references because CoxTime requires smoothness in $g$ and its derivation with respect to $\mathbf{x}$. In Settings 4 and 5, the time grid starts at $t = 2$ because these settings sometimes produce extreme samples with event times as small as $10^{-4}$. In these unrealistic cases, the survival probability drops dramatically from 1 to 0, and the time grid was truncated to exclude such cases.

The studied sample size of the training plus validation data range from $n = 1,000$ to $10,000$, with an 80%-20% split between training and validation. The number of controls varies, being 1, 2, 4, and 8. Each configuration, with $M = 100$ and $B = 200$, is repeated $R = 100$ times. For each repetition $r$, $r = 1, \ldots, R$, survival estimates were evaluated on a new test set of size $n_{test} = 1000$. This resulted in $R = 100$ estimated curves and confidence bands for each test point $\mathbf{x}_i$, $i = 1, \ldots, 1000$. Subsequently, for each test point $i$, we assessed the proportion of times the 100 confidence bands covered the entire true survival curve, based on the adopted grid $\mathcal{T}$. The final reported empirical coverage rate is the average proportion across the $n_{test} = 1000$ individuals. Additionally, to compare different methods, we also examine (half of) the confidence band width, defined by

$$\frac{1}{2n_{test}|\mathcal{T}|} \sum_{i=1}^{n_{test}} \sum_{t \in \mathcal{T}} \{U(t|\mathbf{x}_i) - L(t|\mathbf{x}_i)\}$$

where $|\mathcal{T}|$ is the cardinality of $\mathcal{T}$.

### 3.2 Additional Computational Aspects

The analysis was conducted using the `coxtime` package in Python with Kaiming initialization. We employed a dropout rate of 0.1 and a batch size of 1000. The learning rate was dynamically determined using the `lrfinder` method, and the Adam optimizer was used for optimization. The networks were standard multi-layer DNNs with ReLU activation and batch normalization between layers. In `coxtime`, transformers is utilized, and observed failure times were standardized to have a zero mean and a variance of one, which is necessary due to the inclusion of $t$ in function $g$. We implemented batch normalization with respect to $\mathbf{x}$. The training was conducted for 1500 epochs. To ensure the effectiveness of the proposed ensemble-bootstrap method, it is essential to minimize estimator bias (a standard requirement). Accordingly, for complex settings, we explored various layer configurations, demonstrating that deeper networks with wider layers are necessary in these cases.

For the proposed ensemble-based bootstrap method, each dataset with $n$ training and validation observations had $g(t, \mathbf{x}_i)$, $i = 1, \ldots, n$, estimated $M$ times, resulting in $\widehat{g}_{n,m}(t, \mathbf{x}_i)$, $m = 1, \ldots, M$. We then computed $\widehat{g}_n^M(t, \mathbf{x}_i) = \frac{1}{M} \sum_{m=1}^{M} \widehat{g}_{n,m}(t, \mathbf{x}_i)$ to be used in the Breslow estimator of $H_0(t) = \int_0^t h_0(s)ds$. In particular, define the cumulative hazard $H(t|\mathbf{x}) = \int_0^t h_0(s) \exp\{g(s, \mathbf{x})\}ds$, and its estimator is given by

$$\widehat{H}(t|\mathbf{x}) = \sum_{t_i \leq t} \Delta\widehat{H}_0(t_i) \exp\{\widehat{g}_n^M(t_i, \mathbf{x})\}$$

where

$$\Delta\widehat{H}_0(t_i) = \frac{d_i}{\sum_{j=1}^{n} I(t_j \geq t_i) \exp\{\widehat{g}_n^M(t_i, \mathbf{x}_j)\}} .$$

Subsequently, the ensemble-based survival function estimator is given by $\widehat{S}_n^M(t|\mathbf{x}) = \exp\{-\widehat{H}^M(t|\mathbf{x})\}$.

### 3.3 Results

The simulation results are summarized in Figures 1–4 and include empirical coverage rates and widths of the simultaneous confidence bands of the naive bootstrap method, as well as for the two ensemble-based approaches, one based on Kolmogorov-Smirnov (KS) statistics, $L^{KS}, U^{KS}$, and its modification $L^{prop}, U^{prop}$.

Figures 1 and 2 present the empirical coverage rates and widths for different sizes of controls with $n = 10,000$. The results indicate that the naive bootstrap approach consistently exhibits significant over-coverage across all settings. In contrast, our ensemble approach achieves coverage rates that are reasonably close to the nominal level. Additionally, as the number of controls increases, the coverage rates of the ensemble-based

method improve and their confidence bands become narrower. While larger control sizes might yield better results, they would also be more computationally expensive. Setting 3 demonstrates high sensitivity to the number of layers, with 6 layers and 8 controls providing good performance in terms of coverage rates. Coverage rates in Settings 4 and 5 also improve with an increased number of controls. Our findings suggest that more layers might be required to achieve the "small bias" claim of DNNs survival estimators based on CoxTime of Kvamme et al. (2019). Even though the naive bootstrap procedure can yield over-coverage in cases with highly biased DNN estimates (e.g., Setting 3 with only 4 layers), it should not be regarded as a valid approach. The over-coverage simply indicates that the additional arbitrary noise in the naive bootstrap was large enough to offset the bias.

Figures 3 and 4 present the empirical coverage rates and widths for different sizes of controls with $n = 1000$, 1500 and 2000. Figure 3 reveals that even under small sample sizes, the naive bootstrap approach remains overly conservative, whereas the proposed ensemble-based bootstrap methods perform well in terms of coverage rates. Figure 4 shows that as $n$ increases, the width decreases as expected, and the conservative nature of the naive bootstrap leads to substantially wider confidence bands. Notably, the proportional-KS method consistently produces narrower simultaneous confidence bands while maintaining coverage rates similar to those of the KS method.

Clearly, the ensemble estimator $\widehat{S}_n^M$ exhibits reduced bias and variance compared to that of $\widehat{S}_n$ (results not shown). However, the computational impracticality of constructing a confidence band for the ensemble survival estimator $\widehat{S}_n^M$ persists using the proposed ensemble-based method.

## 4  Simulation Study - Binary Classification

Here we present the results of a smaller simulation study focused on binary classification regression. The outcome, $y_i \in \{0, 1\}$, follows the probability model

$$\Pr(y = 1|\mathbf{x}) = \frac{\exp\{g(\mathbf{x})\}}{1 + \exp\{g(\mathbf{x})\}}$$

where

$$g(\mathbf{x}) = -5 - 4x_1 + 4x_2 + 2x_1 x_2 + x_3^3 + 2\sin(\pi x_4 x_5) + \sum_{j=1}^{10}(-1)^j\{2.5 - 0.25(j-1)\}x_{j+5}\,.$$

The variables $(x_1, x_2)$ were sampled from a bivariate normal distribution with expectation vector $(0, 0)$, unit variances, and a correlation of 0.65. $x_3$ was uniform sampled from $U[-1, 1]$, $x_4$ and $x_5$ were uniformly sampled from $U[0, 1]$, and $x_6, \ldots, x_{15}$ were sampled independently from the standard normal distribution. The objective is to estimate the probability $p(\mathbf{x}) = \Pr(y = 1|\mathbf{x})$ using a DNN estimator along with the corresponding confidence interval based on Eq. (7) and the subsequent algorithm.

We conducted the simulations with sample sizes of $n = 1000, 2000$ and 5000, splitting the data into 80% for training and 20% for validation. Each configuration, with $M = 100$ and $B = 500$, is repeated $R = 100$ times. The test set comprised 1000 observations. The results, shown in Figure 5, were obtained using a DNN with 5, 7 and 9 fully connected hidden layers, each consisting of 128 nodes and utilizing ReLU activation. The output layer consisted of a single neuron with sigmoid activation, and the model was trained using logistic loss. The implementation was done in Keras.

It is apparent that the naive bootstrap approach yields overly conservative coverage rates with unnecessarily wide intervals and the coverage rates are not improved as the number of layers increases. In contrast, the coverage rates of the proposed method are close to the nominal rates and the reduction in confidence interval width achieved by our approach is substantial. Although the proposed method with $n = 1000$ and five layers shows slight undercoverage (e.g., 0.867 instead of 0.90), the coverage rate improves as the number of layers increases (e.g., 0.892 with nine layers). These results align with those observed in Setting 3 of Section 3.1, where the complex function $g$ appears sensitive to the number of layers, and the bias of the DNN estimates tend to decrease substantially as the number of layers increases. Once again, the over-coverage observed in

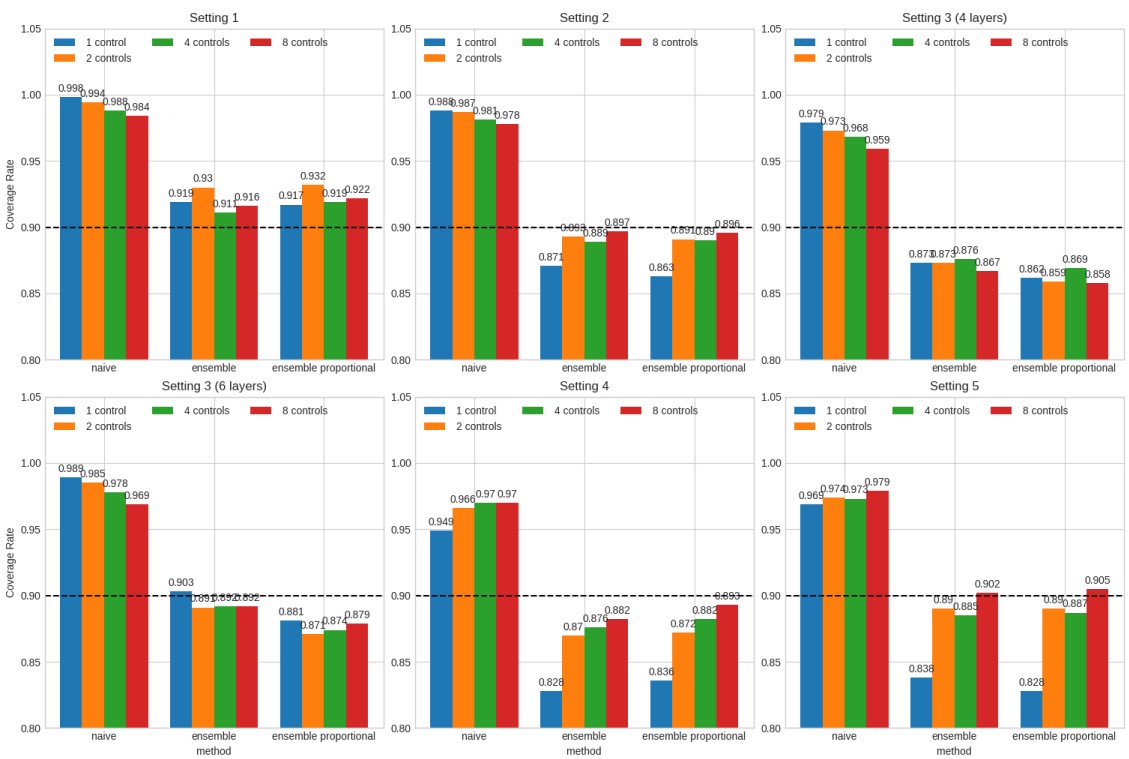

(a) Empirical Coverage Rates of 90% Simultaneous Confidence Band

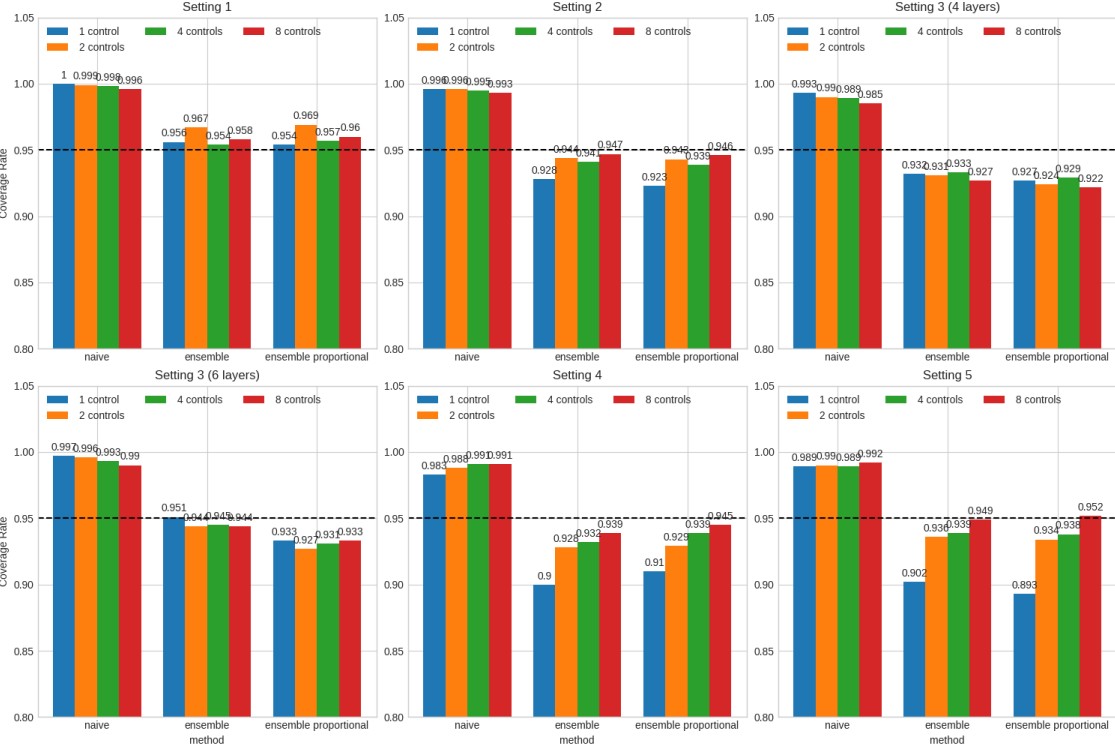

(b) Empirical Coverage Rates of 95% Simultaneous Confidence Band

Figure 1: Simulation results of empirical coverage rates, Settings 1–5 with $n = 10,000$.

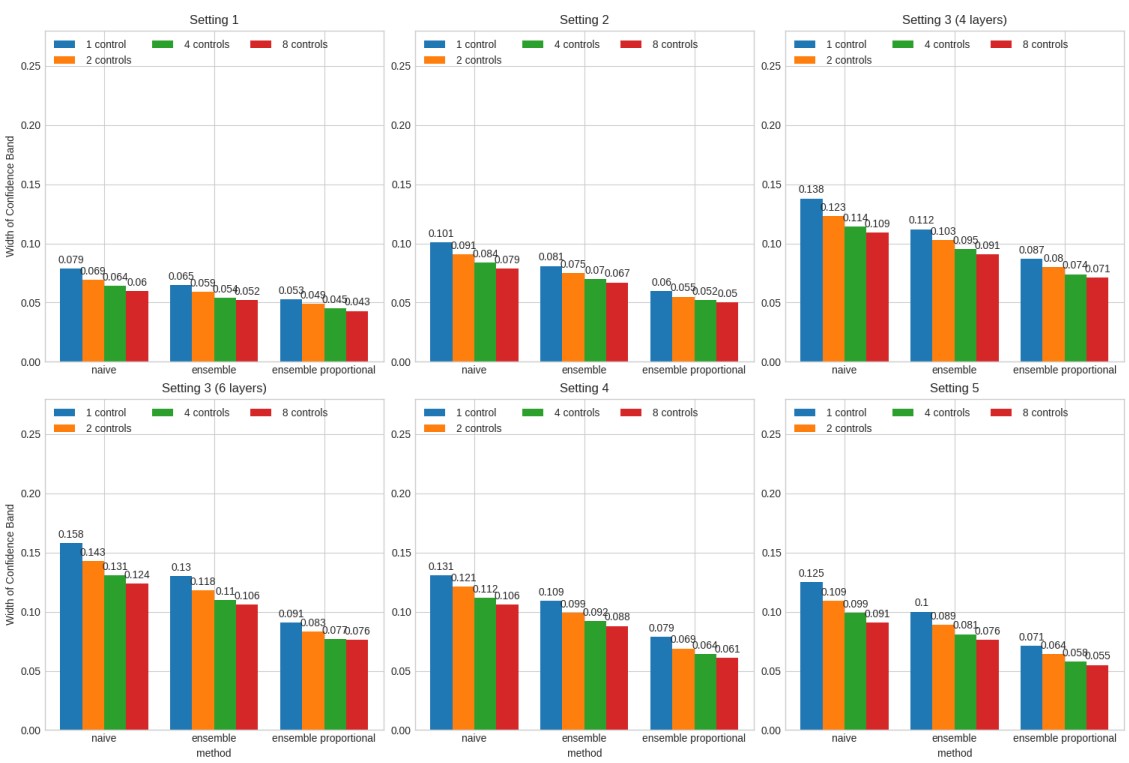

(a) Empirical Width of 90% Simultaneous Confidence Band

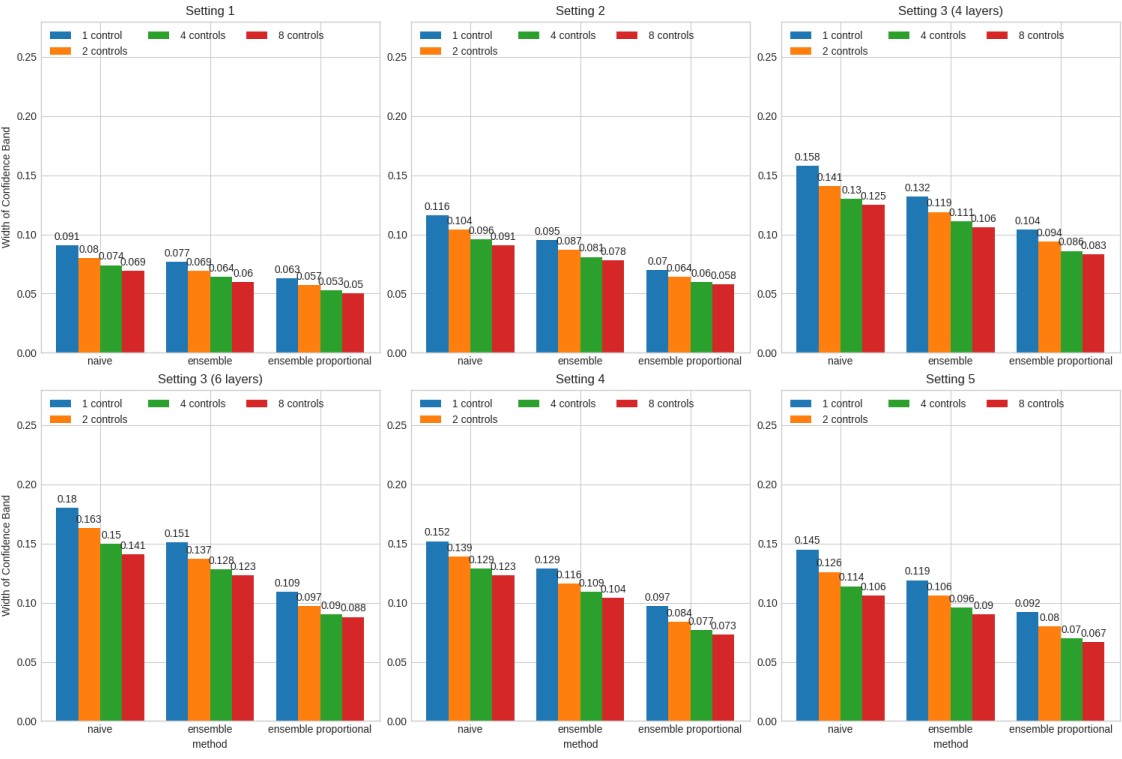

(b) Empirical Width of 95% Simultaneous Confidence Band

Figure 2: Simulation results of empirical width, Settings 1–5 with $n = 10,000$.

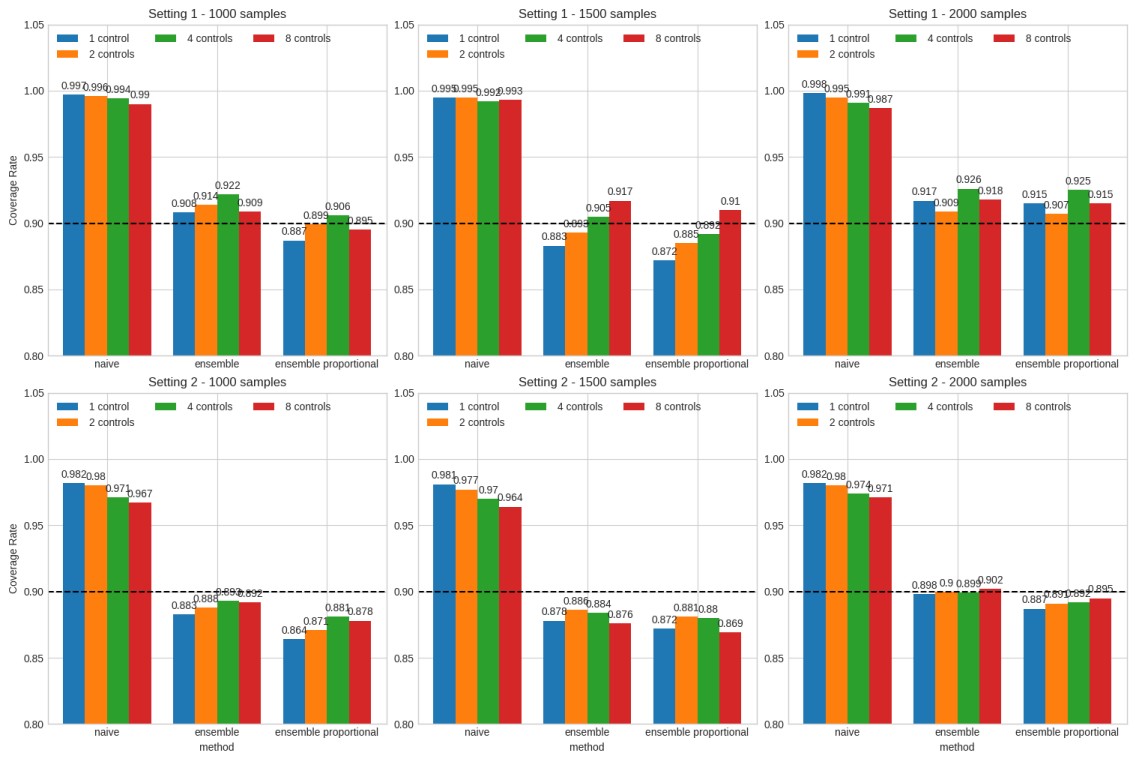

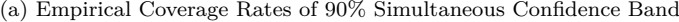

(a) Empirical Coverage Rates of 90% Simultaneous Confidence Band

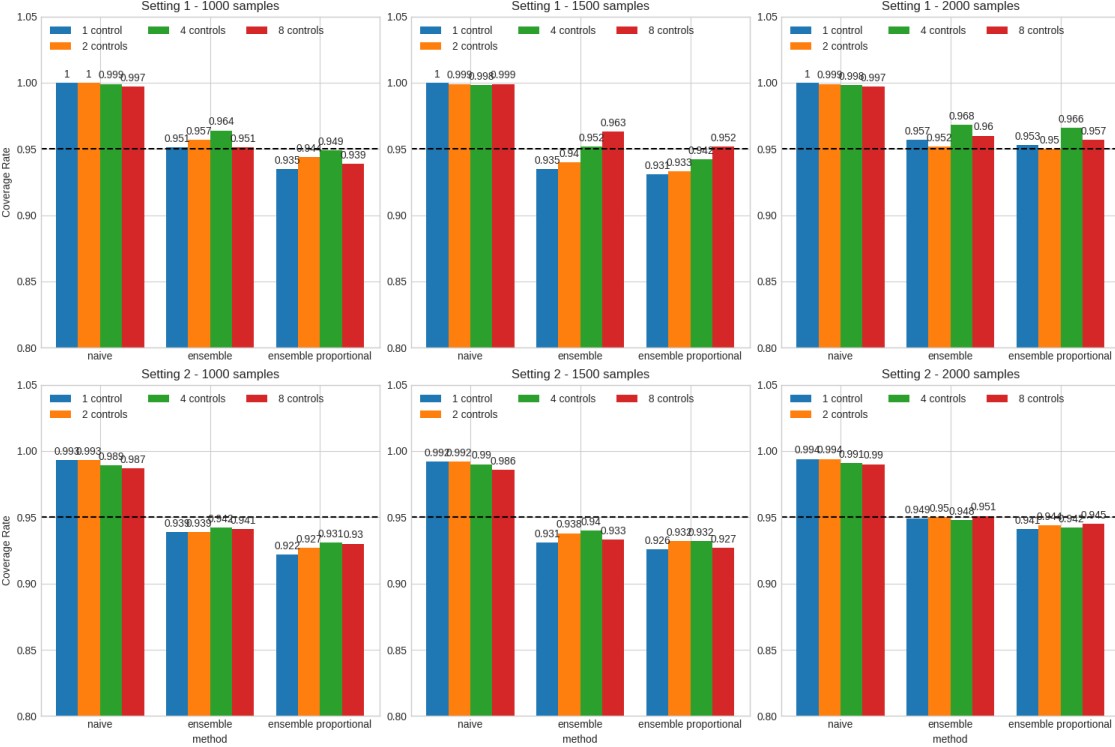

(b) Empirical Coverage Rates of 95% Simultaneous Confidence Band

Figure 3: Simulation results of empirical coverage rates, Settings 1–5 with varied size of training and validation dataset, $n$.

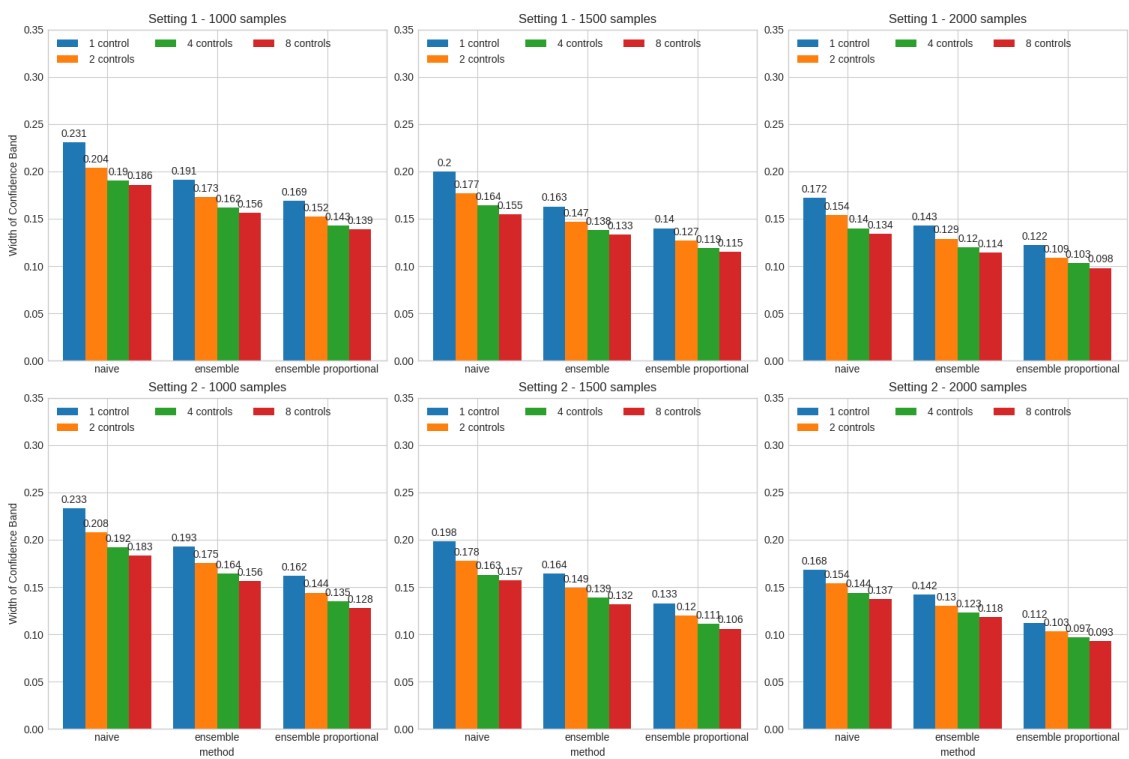

(a) Empirical Mean Width of 90% Simultaneous Confidence Band

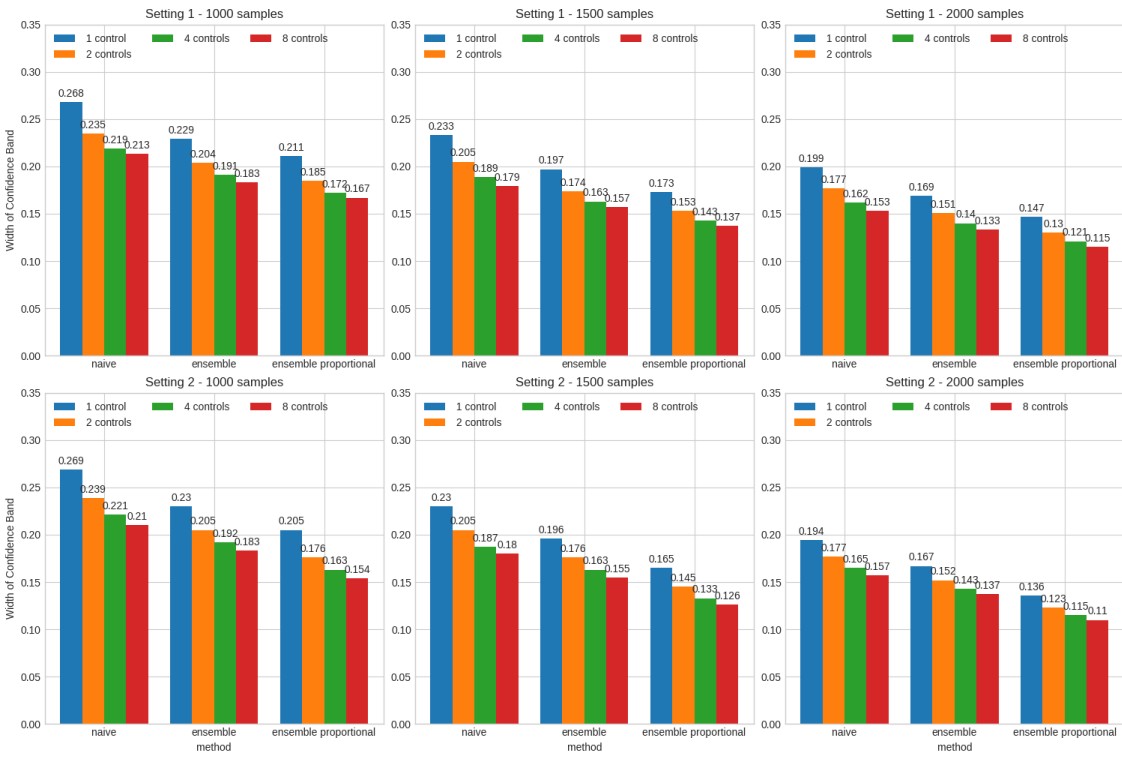

(b) Empirical Mean Width of 95% Simultaneous Confidence Band

Figure 4: Simulation results of empirical mean width, Settings 1–5 with varied size of training and validation dataset, $n$.

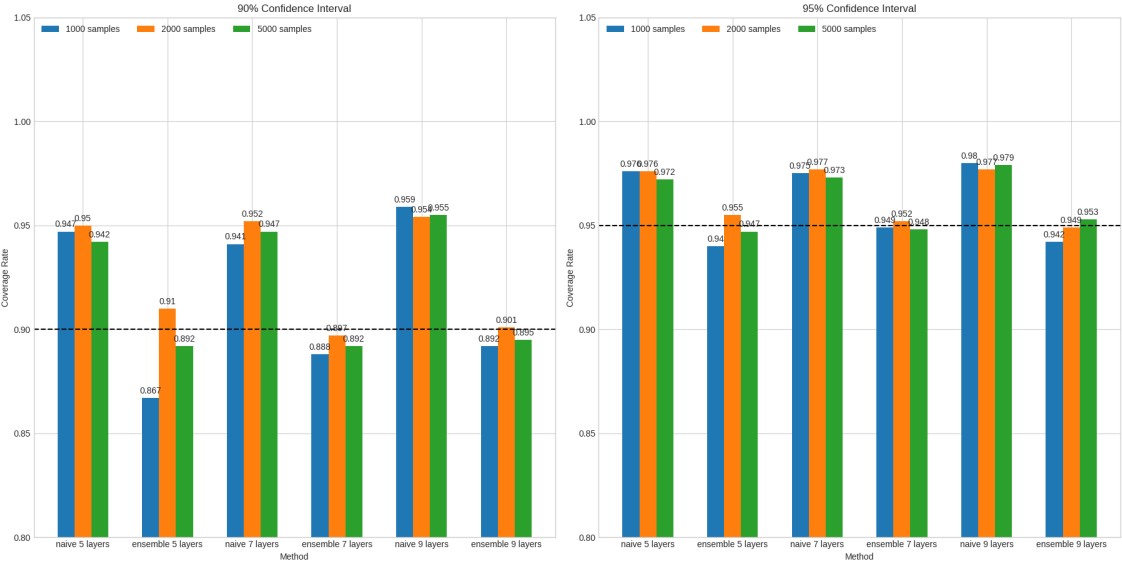

(a) Empirical Coverage Rates of 90% (left) and 95% (right) Confidence Intervals

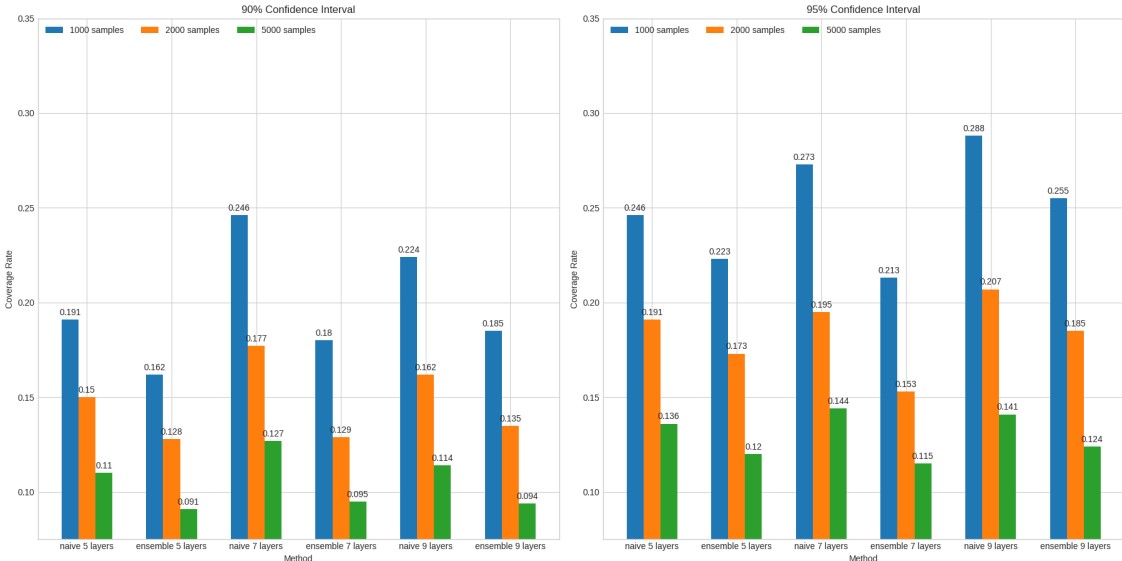

(b) Empirical Mean Width of 90% (left) and 95% (right) Confidence Intervals

Figure 5: Simulation results of binary classification with varied size of training and validation dataset, $n$, and varied number of layers.

the naive bootstrap approach suggests that the added arbitrary noise is sufficient to offset the bias; however, this does not indicate the validity of the approach.

## 5 Experiments

In this section, we analyze four commonly used survival datasets to demonstrate the utility of our proposed approach. These datasets were introduced and used by Katzman et al. (2016) and Kvamme et al. (2019) among others, and are available through `PyCox`. Here are some details of the datasets:

- SUPPORT: Study to Understand Prognoses Preferences Outcomes and Risks of Treatment. This dataset includes 8,873 observations, 14 covariates, and a censoring rate of 0.32.

Table 1: Hyperparameters search space for SUPPORT, METABRIC, Rot. & GBSG and FLCHAIN dataset

| Hyperparameter | Values |
|---|---|
| Learning rate | $10^{-3}$ |
| Patience | 10 |
| Controls | 8 |
| Hidden layers | {1, 2, 4} |
| Layer width | {32, 64, 128, 256} |
| Dropout | {0, 0.1, 0.2} |
| Batch size | {256, 1024} |

Table 2: Mean width of simultaneous confidence bands for the naive and ensemble-based bootstrap methods.

| | Naive | | Ensemble-Based, $M = 100$, $B = 200$ | | | |
| | | | KS | | Prop KS | |
| Dataset | 90% | 95% | 90% | 95% | 90% | 95% |
|---|---|---|---|---|---|---|
| SUPPORT | 0.146 | 0.172 | 0.137 | 0.161 | 0.134 | 0.161 |
| METABRIC | 0.167 | 0.191 | 0.147 | 0.171 | 0.140 | 0.167 |
| Rot.& GBSG | 0.135 | 0.158 | 0.121 | 0.144 | 0.115 | 0.150 |
| FLCHAIN | 0.075 | 0.082 | 0.044 | 0.055 | 0.036 | 0.044 |

- METABRIC: Molecular Taxonomy of Breast Cancer International Consortium. This dataset contains 1,904 observations, 9 covariates, and a censoring rate of 0.42.

- Rot. & GBSG: Rotterdam tumor bank and German Breast Cancer Study Group. This dataset consists of 2,323 observations with 7 covariates and a censoring rate of 0.43.

- FLCHAIN: Assay of Serum Free Light Chain. This dataset includes 6,524 observations with 8 covariates and a censoring rate of 0.70.

Table 2 in Kvamme et al. (2019) shows that CoxTime and DeepSurv are the top two methods based on the concordance measure when analyzing these datasets. In the current analysis, pre-encoded categorical variables were used as-is, while numerical variables were standardized following Kvamme et al. (2019)'s recommendations. Hyperparameters tuning was conducted using cross-validation. A grid search over the hyperparameters search space, as detailed in Table 1, was performed by splitting the data into 10 folds for each configuration and scoring the C-index on the held-out set. The set of hyperparameters with the highest average C-index was selected. The following are the selected hyperparameters for each dataset: SUPPORT - 4 hidden layer, layer width of 256, dropout 0.3 and batch size of 256; METABRIC - 2 hidden layer, layer width of 256, dropout 0.3 and batch size of 1024; Rot.&GBSG - 1 hidden layer, layer width of 128, dropout 0.1 and batch size of 256; and FLCHAIN - 1 hidden layer, layer width of 256, dropout 0.1 and batch size of 256.

The results of the confidence bands analysis, summarized in Table 2, are based on 10-fold cross-validation. The held-out fold serves as the test set, while the remaining data were split 80%-20% into training and validation sets. The total computation time required for each dataset, including the selection of hyperparameters, is as follows: SUPPORT - 210 minutes, METABRIC - 66 minutes, Rot.& GBSG - 52 minutes, and FLCHAIN - 98 minutes. Examples of survival curves along with their confidence bands can be found in Figure 6. As anticipated, the widths of the ensemble-based methods are shorter than those of the naive bootstrap, with the proportional-KS method generally having the narrowest widths.

## 6 Theoretical Aspects

The following discussion is divided into two parts: (i) The motivation for the proposed bootstrap method for estimating the uncertainty of $\widehat{\mu}_n$, based on the assumption that $\widehat{\mu}_n$ is a consistent estimator of $\mu^o$ and

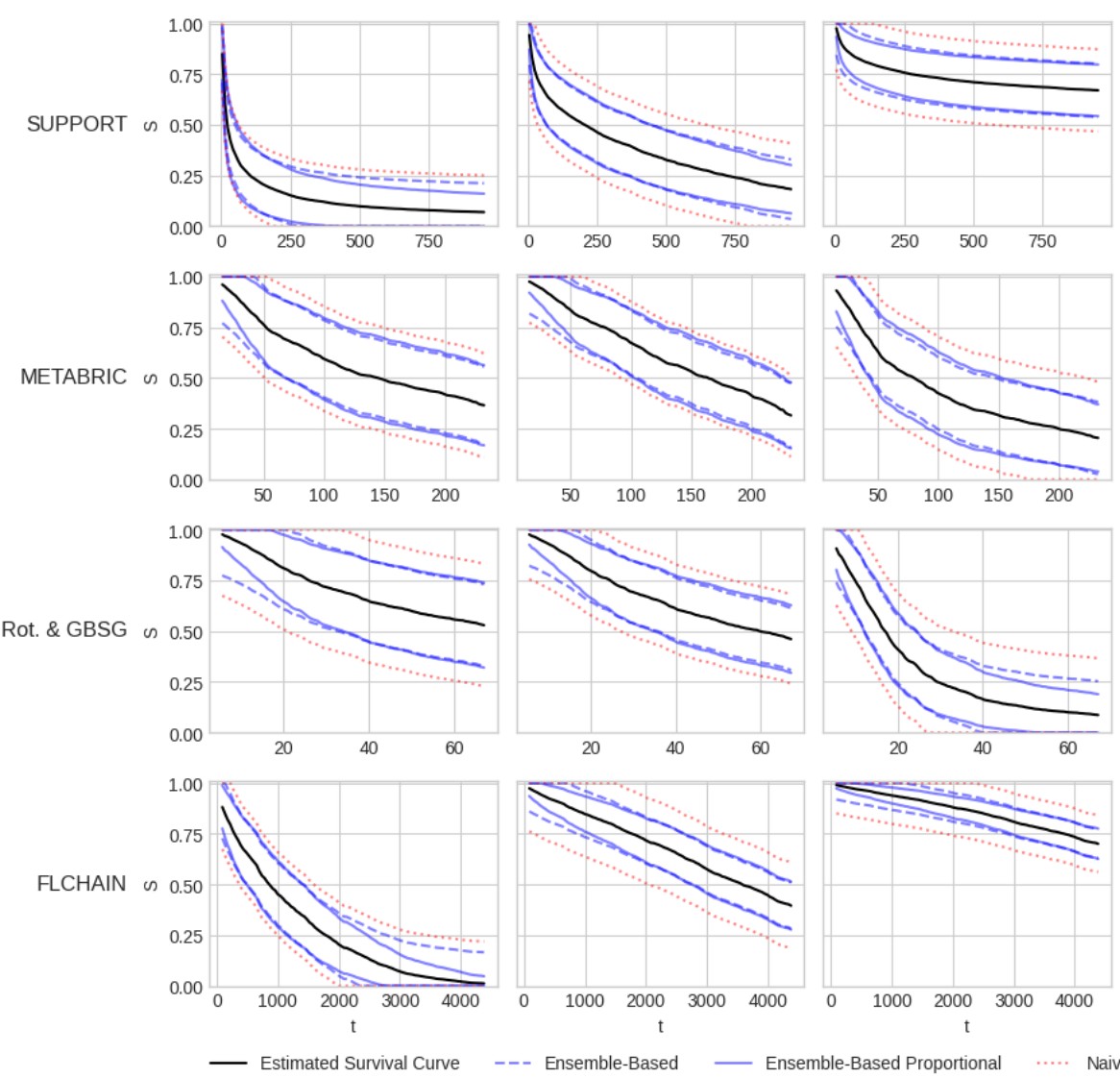

Figure 6: Examples of survival curves and simultaneous confidence bands produced by the naive and ensemble-based bootstrap methods.

the decomposition of $\widehat{\mu}_n$ and $\widehat{\mu}_n^{(b)}$, as defined in (4) and (5), hold true. (ii) A discussion on existing deep neural networks that produce consistent estimators.

Assume that $M$ is sufficiently large such that the uncertainty of the ensemble estimator due to the inherent randomness of the DNN optimization process, is zero; that is, $var\left(\widehat{\mu}^M\right) = v_1$ and $var\left(\widehat{\mu}^M\,|\,\mathcal{D}_n\right) = 0$. For simplicity, we focus on the asymptotic variance, and similar arguments can be used for other statistical functionals. We write,

$$
\begin{aligned}
var(\widehat{\mu}_n) &= E\left\{\left(\widehat{\mu}_n - \widehat{\mu}^M + \widehat{\mu}^M - \mu^o\right)^2\right\} \\
&= E\left\{\left(\widehat{\mu}_n - \widehat{\mu}^M\right)^2\right\} + E\left\{\left(\widehat{\mu}^M - \mu^o\right)^2\right\} + 2E\left\{\left(\widehat{\mu}_n - \widehat{\mu}^M\right)\left(\widehat{\mu}^M - \mu^o\right)\right\} \\
&= E\left\{(Z_1 + Z_2 - Z_1)^2\right\} + E\{(Z_1 - \mu^o)^2\} + 2E\left\{(Z_1 + Z_2 - Z_1)(Z_1 - \mu^o)\right\} \\
&= var\left(Z_2\right) + var\left(Z_1\right) + 2cov\left(Z_2, Z_1\right) \\
&= v_1 + v_2 + 2v_{12}\,.
\end{aligned}
$$

The third equality holds based on the main concept in Section 2.2, which decomposes $\widehat{\mu}_n$ into $Z_1 + Z_2$, where $Z_1$ is fixed given $\mathcal{D}_n$, and $Z_2$ represents the additional variability introduced by the DNN optimization process. While the variance of $Z_2$ is not zero given $\mathcal{D}_n$, the ensemble estimator, with sufficiently large $M$, ensures that $Z_2 \equiv 0$.

In a scenario with unlimited computational resources, one would ideally base predictions or estimations on $\widehat{\mu}_n^M$ instead of $\widehat{\mu}_n$. The corresponding bootstrap samples, denoted by $\widehat{\mu}_n^{M(b)}$, $b = 1, \ldots, B$, could then be used for uncertainty estimation, with $var\left(\widehat{\mu}_n^{M(b)}|\mathcal{D}_n\right) = v_1$. However, since $\widehat{\mu}_n$ is used in practice, we now turn our attention to the recommended approach for variance estimation, namely,

$$
\begin{aligned}
E\left\{\left(\widehat{\mu}_n^{(b)} - \widehat{\mu}_n^M\right)^2 |\mathcal{D}_n\right\} &= E\left\{\left(\widehat{\mu}_n^{(b)} - \widehat{\mu}_n^{M(b)} + \widehat{\mu}_n^{M(b)} - \widehat{\mu}_n^M\right)^2 |\mathcal{D}_n\right\} \\
&= E\left\{\left(\widehat{\mu}_n^{(b)} - \widehat{\mu}_n^{M(b)}\right)^2 |\mathcal{D}_n\right\} + E\left\{\left(\widehat{\mu}_n^{M(b)} - \widehat{\mu}_n^M\right)^2 |\mathcal{D}_n\right\} \\
&\quad + 2E\left\{\left(\widehat{\mu}_n^{(b)} - \widehat{\mu}_n^{M(b)}\right)\left(\widehat{\mu}_n^{M(b)} - \widehat{\mu}_n^M\right) |\mathcal{D}_n\right\} \\
&= E\left\{\left(Z_1^{(b)} + Z_2^{(b)} - Z_1^{(b)}\right)^2 |\mathcal{D}_n\right\} + E\left\{\left(Z_1^{(b)} - \widehat{\mu}_n^M\right)^2 |\mathcal{D}_n\right\} + 2cov(Z_1^{(b)}, Z_2^{(b)}|\mathcal{D}_n) \\
&= v_1 + v_2 + 2v_{12}\,.
\end{aligned}
$$

The third equality follows from similar reasoning as before, specifically by decomposing $\widehat{\mu}_n^{(b)}$ into $Z_1^{(b)}$ and $Z_2^{(b)}$. For $\widehat{\mu}_n^{M(b)}$ (which does not need to be computed in practice), $Z_2^{(b)} \equiv 0$. The last equality assumes that $var(Z_1) = var(Z_1^{(b)}) = v_1$, $var(Z_2) = var(Z_2^{(b)}) = v_2$, and $cov(Z_1, Z_2) = cov(Z_1^{(b)}, Z_2^{(b)}) = v_{12}$. These assumptions are based on a common principle (or proven result) in bootstrap methods, often referred to as bootstrap distribution consistency. In our setting, this principle asserts that the (asymptotic) distribution of $a(n)\left(\widehat{\mu}_n - \mu^o\right)$ is equal to the (asymptotic) conditional distribution of $a(n)\left(\widehat{\mu}_n^{(b)} - \widehat{\mu}_n^M\right)$, given $\mathcal{D}_n$, where $a(n)$ is typically $\sqrt{n}$. It is important to note that, due to the DNN optimization error, we use $a(n)\left(\widehat{\mu}_n^{(b)} - \widehat{\mu}_n^M\right)$ instead of $a(n)\left(\widehat{\mu}_n^{(b)} - \widehat{\mu}_n\right)$.

Under mild regularity conditions, and using a DNN whose complexity grows with the data, as often used in practical applications, Schmidt-Hieber (2020) established consistency and convergence rate of the nonparametric function estimator in the regression model $Y = g(\mathbf{x}) + \epsilon$ where $g$ is an unknown function and $\epsilon$ is a random noise. Recently, Zhong et al. (2022) extended these results to the partially linear Cox mode of the form $h_0(t) \exp\{\theta^T \mathbf{z} + g(\mathbf{x})\}$. The work of Zhong et al. (2022) covers Settings 1,2,4 and 5 of our numerical study, but do not cover Setting 3 with $g(t, \mathbf{x})$, although our numerical results are excellent when the number of layers is large enough.

## 7   Concluding Remarks

In this paper, we introduced a bootstrap methodology to estimate uncertainty in predictions or estimations based on DNN. The core concept involves an ensemble approach to separate data uncertainty from the noise inherent in the optimization algorithm. Our method is general and can be applied to any DNN analysis without interfering with or compromising the DNN predictions themselves.

We demonstrated the utility of our general approach in the context of survival analysis, where the primary focus is on estimating the survival function based on a new set of features and in the context of binary classification. Our comprehensive simulation study indicates that the proposed method is valid and not overly conservative, as long as the estimators provided by the DNN have only a small bias. Otherwise, undercoverage of up to 8% is observed. Often, bias reduction in DNN can be achieved by increasing the number of layers and nodes. Future research should aim to refine the proposed approach to reduce its computational complexity and perform a comprehensive theoretical analysis to rigorously establish the (asymptotic) accuracy of the coverage probability across diverse conditions. Code for the data analysis and reported simulation is avalable at Githubsite: `https://github.com/Asafba123/Survival_bootstrap`.

## Acknowledgments

We thank the anonymous reviewers for helpful discussions and feedback. The work was supported by the Israel Science Foundation (ISF) grant number 767/21 and by a grant from the Tel-Aviv University Center for AI and Data Science (TAD).

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
