# OpenReview forum: "Confidence Intervals and Simultaneous Confidence Bands Based on Deep Learning"
_TMLR — Accepted by TMLR_

### Review · Reviewer_AGt3 · 2024-07-22

**Summary Of Contributions:**

This paper presents a bootstrap approach for constructing prediction intervals, specifically focusing on survival analysis.

The authors offer two new bootstrap variants, which are shown empirically to be less conservative than the naive one. In particular, the truncated variant is designed to avoid instabilities involved in the normalization term.

Numerical experiments support the approximate validity of the proposed methods.

**Audience:**

Yes

**Broader Impact Concerns:**

I don't see an issue here.

**Claims And Evidence:**

Yes

**Requested Changes:**

I strongly encourage the authors to support their methods with theoretical analysis for the coverage guarantee. This is of great importance given the high-risk nature of the problems of interest. Notably, the theoretical analysis can even be done for special cases and under simplifying assumptions. The goal here is to shed light on the correctness of the proposed algorithms, as a way to further support the empirical results.

**Strengths And Weaknesses:**

**Strengths**

+ Well-written paper.
+ The advantage of the proposed methods is supported by experiments.
+ Experiments highlight the usefulness of the proposed methods in practical settings.

**Weaknesses**

- My main concern is the lack of theoretical analysis; see the requested changes.
- Numerical experiments show cases where the lower bound is too liberal.

---

> ### Author Response · Authors · 2024-08-29
> **We thank you for the careful review and thoughtful comments about our paper. Below is our response to your comment.**
>
> We have added Section 6 which includes a discussion and an outline of a proof under certain conditions. We believe that a full proof would warrant  a separate paper.

---

> ### Comment · Reviewer_AGt3 · 2024-09-18
>
> Thank you for your reply. I found that the discussion in Section 6 requires some clarification.
>
> First, I don’t fully understand why it is sensible to assume that $v_1$, $v_2$, and $v_{12}$ are the same when analyzing $\hat{\mu}_n$ and $\hat{\mu}_n^b$.
>
> Second, in the derivation of $var(\hat\mu_{n})$, I recommend providing an explanation for the third equality where $Z_1$, $Z_2$, and $Z_{12}$ are introduced. The same applies to the derivation of $var(\hat\mu_n^{(b)} - \hat \mu_n^M)$; in this case, it's also the third equality that could benefit from clarification.
>
> In sum, the idea of the paper is interesting, however, numerical evidence shows that the proposed method can *undercover* the target variable in contrast with the naive bootstrap method that usually overcovers the target. I believe the undercoverage issue requires further explanation and analysis---it is crucial to better understand when to expect the proposed method to fail.

---

> > ### Author Response · Authors · 2024-10-07
> > **We thank you for the additional thoughtful comments. Below is our response.**
> >
> > 1) We revise Section 6 while taking into account all your comments:
> >
> > 1.1) In particular, we explain that it is assumed that $\text{var}(Z_1) = \text{var}(Z_1^{(b)}) = v_1$, $\text{var}(Z_2) = \text{var}(Z_2^{(b)}) = v_2$, and $\text{cov}(Z_1, Z_2) = \text{cov}(Z_1^{(b)}, Z_2^{(b)}) = v_{12}$. These assumptions are based on a common principle (or proven result) in bootstrap methods, often referred to as bootstrap distribution consistency. In our setting, this principle asserts that the (asymptotic) distribution of $a(n) \left( \widehat{\mu}_n - \mu^o \right)$ is equal to the (asymptotic) conditional distribution of $a(n) \left( \widehat{\mu}_n^{(b)} - \widehat{\mu}_n^M \right)$, given $\mathcal{D}_n$, where $a(n)$ is typically $\sqrt{n}$. It is important to note that, due to the DNN optimization error, we use $a(n) \left( \widehat{\mu}_n^{(b)} - \widehat{\mu}_n^M \right)$ instead of $a(n) \left( \widehat{\mu}_n^{(b)} - \widehat{\mu}_n \right)$.
> >
> > 1.2) In addition, the third equality re $var(\widehat{\mu}_n)$ holds based on the main concept in Section 2.2, which decomposes $\widehat{\mu}_n$ into $Z_1 + Z_2$, where $Z_1$ is fixed given $\mathcal{D}_n$, and $Z_2$ represents the additional variability introduced by the DNN optimization process. While the variance of $Z_2$ is not zero given $\mathcal{D}_n$, the ensemble estimator, with sufficiently large $M$, ensures that $Z_2 \equiv 0$. A similar reasoning holds for the bootstrap variance, and this is now noted in the text.
> >
> > 2) The simulation study of the binary classification setting has been expanded to include also 7 and 9 layers, in contrast to only 5 layers. It is apparent that the naive bootstrap approach yields overly conservative coverage rates with unnecessarily wide intervals and the coverage rates are not improved as the number of layers increases. In contrast, the coverage rates of the proposed method are close to the nominal rates and the reduction in confidence interval width achieved by our approach is substantial. Although the proposed method with $n=1000$ and five layers shows slight undercoverage (e.g., 0.867 instead of 0.90), the coverage rate improves as the number of layers increases (e.g., 0.892 with nine layers). These results align with those observed in Setting 3 of Section 3.1, where the complex function $g$ appears sensitive to the number of layers, and the bias tends to decrease as the number of layers increases.

---

### Review · Reviewer_m1E5 · 2024-08-16

**Summary Of Contributions:**

The paper  focuses on addressing the challenge of predicting uncertainty in deep learning models, particularly for survival data. The authors propose a non-parametric bootstrap method to construct valid, non-overly conservative point-wise confidence intervals along with simultaneous confidence bands. This method can be integrated into any deep neural network (DNN) without affecting the training process. The paper demonstrates the practicality of this approach using survival curves derived from DNNs handling right-censored survival data.

**Audience:**

Yes

**Broader Impact Concerns:**

The paper's work on uncertainty quantification in deep learning, particularly for survival analysis, raises numerous ethical concerns. These include potential misuse in critical fields such as medicine, where inaccuracies could affect patient care, as well as the risk of biased predictions unfairly impacting certain groups.

**Claims And Evidence:**

Yes

**Requested Changes:**

First, the evaluation should be expanded across diverse DNN architectures to demonstrate the method's generalizability beyond survival analysis, which is crucial for broader applicability.

Second, the discussion on the underlying assumptions of the proposed method needs to be strengthened to clarify how these assumptions impact the validity of the results, ensuring that readers understand the method's limitations.

Last, optimizing the computational complexity of the method is essential, as the current high computational cost may limit its practical use, especially in large-scale applications.

**Strengths And Weaknesses:**

**Strengths**

1.	The paper introduces a novel bootstrap method that effectively separates data uncertainty from algorithmic noise, offering a significant advancement in the field of uncertainty quantification for deep learning models.

2.	The proposed method is designed to be easily integrated into existing DNN frameworks without requiring changes to the training process, making it highly practical.

3.	The method ensures that the produced confidence intervals and bands are accurate and not overly conservative, providing more reliable uncertainty estimates compared to existing approaches.

4.	The authors provide extensive validation through simulation studies and application to real-world survival data, demonstrating the robustness and effectiveness of their approach.

**Weaknesses**

1.	The ensemble-based bootstrap method requires running multiple DNNs, which can be computationally expensive and time-consuming, potentially limiting its scalability to extremely large datasets.

2.	The validity of the proposed method relies on certain assumptions regarding DNN estimators, which may not always hold in practice, affecting the generalizability of the results.

3.	Although the method is theoretically applicable to various DNNs, the paper predominantly focuses on survival analysis, with limited demonstration across other types of deep learning applications.

4.	The empirical evaluation, though extensive, should be expanded by including a wider range of DNN architectures to further validate the method's versatility and effectiveness across different scenarios.

---

> ### Author Response · Authors · 2024-08-29
> **We thank you for the careful review and thoughtful comments about our paper. Below is our  response to your comments.**
>
> 1) The new Section 4 of the revised paper provides a simulation study for the setting of binary classification regression. The results show that the naive bootstrap approach yields overly conservative coverage rates with unnecessarily
> wide intervals. In contrast, the coverage rates of the proposed approach are close to the nominal rates, provided
> the sample size is sufficiently large. Additionally, the reduction in confidence interval width achieved by our
> approach is substantial.
>
> 2) Thank you for your comment. We have added Section 6 which includes a discussion.
>
> 3) We agree that very large datasets would require substantial computational resources and efficient parallel computing to implement our approach. We have acknowledged this issue in Section 5 and noted that future research should aim to enhance the proposed method to reduce computational demands. Meanwhile, we demonstrated that analyzing a dataset with 8,873 observations took a total computation time of 210 minutes.

---

> > ### Comment · Reviewer_m1E5 · 2024-09-20
> >
> > Thanks for the detailed response and the empirical evidence provided on coverage rates and confidence interval widths. This information significantly enhances the understanding of the practical benefits of proposed method. Additionally, the inclusion of Section 6 further solidifies the approach and provides a more comprehensive framework.
> >
> > In conclusion, the paper presents an interesting idea and is well articulated. However, I would like to reiterate the importance of continuing efforts to reduce the computational demands of the method. Doing so would greatly enhance its practical applicability.

---

### Review · Reviewer_R19s · 2024-08-21

**Summary Of Contributions:**

The paper proposes a new approach for uncertainty quantification in deep learning using a non-parametric bootstrap approach where the bootstrap distribution is evaluated around an ensemble estimator of the prediction at a point to eliminate the uncertainty due to the training procedure. The estimated confidence intervals can also be augmented with a variable width component based on an estimate of the standard error of the predictor which. The authors evaluate the approach specifically for the case of survival analysis where the data is typically censored which adds an additional challenge to the task of uncertainty quantification. The proposed estimators generally give good coverage rates with narrower intervals than the baseline bootstrap estimators.

**Audience:**

Yes

**Claims And Evidence:**

Yes

**Requested Changes:**

1. Why do you say that current methods for prediction uncertainty are not applicable for survival data with censoring (last line of second paragraph in Section 1.3)?

2. Is it wrong to use the same initialization, hyperparameters and validation set for each bootstrap sample? If not, then wouldn't that eliminate the variability due to the training process?

3. Keeping aside the concern around computational cost, why not apply M repetitions i.e. ensembling to the bootstrap estimates also? Would that not further reduce the uncertainty?

4. Why are the plots for settings 3-5 missing in Fig 3 and 4?

**Strengths And Weaknesses:**

Strengths:

1. The approach is well motivated and the problem of variability due to randomness in training as a cause of overly conservative confidence intervals does not appear to have been considered before.

2. The approach empirically outperforms the baselines in terms of more precise coverage with narrower confidence intervals for a range of simulation settings and real-world datasets.

Weaknesses:

1. The proposed approach increases the computational cost of inference since it requires at least M extra DNN runs (due to ensembling). However, the authors have acknowledged this and recommended that future work should look into it and so I don't think it is a major concern at this stage.

2. While the authors claim to provide exactly 1-$\alpha$ coverage in Section 2.3 there is no proof for this. Moreover, results in Section 3 show that the coverage is never exactly 1 -$\alpha$ (corresponding to 90% in Fig 1.a and 95% in Fig 1.b). Thus, I feel that it should be clearly stated in section 2.3 that such exact coverage cannot be guaranteed.

---

> ### Author Response · Authors · 2024-08-29
> **We thank you for the careful review and thoughtful comments about our paper. Below is our point-by-point response to your comments.**
>
> 1) Existing deep learning approaches for survival analysis typically do not incorporate uncertainty estimation, particularly for the estimated survival curves. Current methods for uncertainty estimation in deep learning without censored data are either limited to specific deep neural networks and often compromise the training procedure, or they can be applied to a variety of deep neural networks but rely on residuals as one of their core components, as seen in the discriminative jackknife method by Alaa & van der Schaar (2020). However, residuals are not fully observed in survival data due to right censoring. Section 1.1 provides an overview of existing works on uncertainty quantification in deep learning. Furthermore, the approach by Alaa & van der Schaar (2020) provides a prediction interval for the outcome of a new observation. However, in survival analysis, for example, the primary interest often lies in estimating the survival function for a new observation, as discussed in Section 1.2.
>
> 2) Our proposed approach involves using the same hyperparameters and validation set across the bootstrap samples. If one begins by drawing the bootstrap sample first and then splitting it into training and validation sets, it can create an overlap between these sets that is not present in the original training and validation sets. This results in an underestimated uncertainty. Therefore, we first split the data into training and validation sets, and bootstrap samples are generated only from the training data.
> Using fixed initial values in deep learning is generally not recommended for several reasons (e.g., symmetry breaking, poor convergence and vanishing or exploding gradients). To address these issues, common practice involves using random initialization strategies (Glorot & Bengio, 2010; He et al., 2015; LeCun et al., 2015), and the same approach should be adopted when applying the analysis to bootstrap datastes.
> Moreover, most deep learning methods incorporate additional randomness within the optimization process (see Section 3 of the paper), which can degrade performance if fixed throughout the analysis. In general, fixing the random components introduced by the optimization algorithm can be impossible, such as with batch splitting, because bootstrap samples are not identical. Even in cases where it is feasible in deep learning (and does not harm or compromise the optimization procedure), such bootstrap samples only capture the noise from the observed data while ignoring the uncertainty introduced by the optimization algorithm itself, leading to an underestimation of overall uncertainty.
> For all these reasons, we believe it is crucial to develop an uncertainty estimator that can effectively account for any randomness in the optimization steps of deep neural networks, without compromising their performance.
>
> 3) To reduce uncertainty, one should focus on improving the prediction or estimator derived from the original data. One approach is to use an ensemble estimator and then apply the bootstrap method to this ensemble estimator. However, as the reviewer noted, this approach is impractical due to the significant computational burden it imposes.
>
> 4) The aim of Figures 3 and 4 was to also demonstrate the effect of sample size, so configurations were studied with $n=1000, 1500$ and 2000. To save computation time, we conducted this analysis only for Settings 1 and 2. We need more time to provide additional simulation results for Settings 3-5.
>
> 5) We agree that there is currently no general proof, and the necessary assumptions should be discussed in greater detail. We have added Section 6 which includes a discussion. The wording in Section 2.3 has been revised accordingly.

---

### Decision · Action_Editor_MeJN · 2024-10-19

**Recommendation:** Accept with minor revision

**Comment:**

This paper proposes a novel non-parametric bootstrap method for computing confidence intervals and simultaneous confidence bands for deep learning models, focusing on survival analysis.

The reviewers agree that the paper is well-written, engaging, and well-motivated. However, the proposed method lacks theoretical guarantees, and the empirical results show that the method's proposed confidence intervals undercover the target variable.

Thus, to make the paper fully evidenced for the camera-ready, the authors should clarify in the abstract and throughout the paper that they focus on empirical results, acknowledge that the empirical undercovers the target variable (up to 8% points in some settings?), and only provide a theoretical motivation for the method. The statement that the method provides valid and accurate confidence intervals (as stated in the abstract) must be fixed and qualified.

Usually, uncertainty quantification methods are conservative (like the used baseline), as it is better to overcover than undercover in many settings, such as when there are safety risks or in medical settings.

Overall, this is an acceptance with minor revisions, and I will closely follow up with the authors to ensure that the changes are incorporated.

**Audience:**

The reviewers agree that the paper is interesting. It provides a non-parametric bootstrap method to compute narrower confidence intervals using deep learning models (with a focus on survival analysis).

**Claims And Evidence:**

Yes and no. Overall, the claims made in the submission are supported. The authors provide theoretical motivation and empirical results from synthetic and real-world settings. At the same, the empirical experiments show that the method's confidence intervals undercover the target variable (while being narrower).

AE note: Indeed, the naive baseline seems to have less variance in how much it overcovers the target variable compared to this method.

---

> ### Author Response · Authors · 2024-10-31
>
> We appreciate your constructive comments and are grateful for the decision of 'Accept with Minor Revision.'
>
> The revised sections of the paper are highlighted in red. We have revised the paper throughout to highlight the following points:
>
> 1. The proposed approach is supported by empirical validation.
>
> 2. Generally, confidence interval or simultaneous confidence band methods based on point estimators require 'small bias' in the estimators to be valid. In our simulation study involving a complex function $g$, we demonstrate that achieving minimal bias, and thus the desired coverage rate, requires a sufficiently deep DNN. For instance, in Setting 3, the simulation results with a 4-layer network (Figure 1(a)) show that 4 layers (even with 8 controls) are insufficient, while a 6-layer network with 8 controls achieves good results. These findings do not invalidate our proposed approach but rather highlight the importance of tuning the DNN parameters to minimize bias effectively.
>
> 3. Although the simulation study shows that the naive bootstrap procedure provides conservative confidence intervals or simultaneous confidence bands, it should not generally be regarded as a valid approach, as it includes \underscore{\bf{arbitrary}} noise that does not reflect the true uncertainty we aim to estimate. This is demonstrated and discussed in detail in the simulation study in Sections 3 and 4.

---

> ### Comment · Action_Editor_MeJN · 2024-11-20
> **I cannot certify camera ready as is.**
>
> The camera-ready currently does not contain the changes I requested as part of the decision notice. In particular, the acknowledgment of the evidence that the proposed method significantly undercovers in many of the experiments (see decision notice for details) and the lack of theoretical proof/focus on empirical evaluations.
>
> Hence, I would suggest either updating the camera-ready accordingly or allowing the EiC to reject the paper for now and enable the authors to update the paper more significantly to evidence the current claims.
>
> Thanks,\
> Andreas

---

> > ### Comment · Action_Editor_MeJN · 2024-11-24
> > **Thank you!**
> >
> > I thank the authors for taking my previous feedback into account and am happy to certify the camera-ready revision of the paper now.
> >
> > I believe these minor edits improve the paper overall, and I'm looking forward to reading an expanded theoretical treatment of the suggested method in the future.
> >
> > Best wishes,\
> > Andreas